# SEED-X: MULTIMODAL MODELS IN REAL WORLD

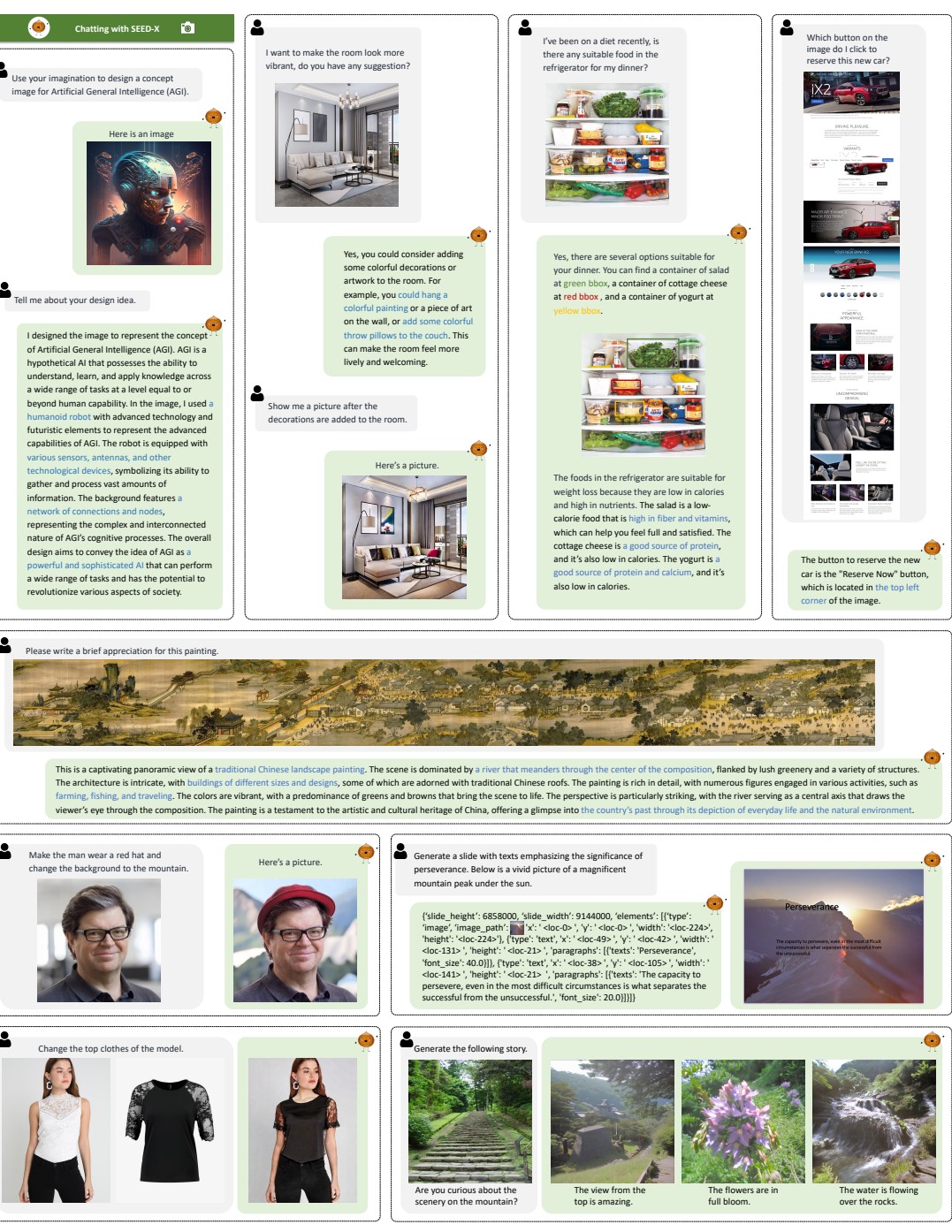

Figure 1: The introduced SEED-X, a unified and versatile foundation model, can serve as multimodal AI assistants **in the real world** after instruction tuning, capable of responding to a variety of user needs through unifying **multi-granularity comprehension and generation**.

ABSTRACT

The rapid evolution of multimodal foundation models has showcased remarkable capabilities in vision-language understanding and generation, yielding impressive results on academic benchmarks. However, there remains a gap in their progress toward real-world applicability, primarily due to the models' limited capacity to effectively respond to various user instructions and interact with diverse visual data. This limitation can be attributed to the fundamental challenge of modeling multi-granularity visual semantics for comprehension and generation tasks. In this paper, we take a pioneering step towards applying multimodal foundation models in an open-world context and present a unified and versatile foundation model, namely, **SEED-X**. As the first of its kind, SEED-X seamlessly integrates two essential features: (1) comprehending images of arbitrary sizes and ratios, and (2) enabling multi-granularity image generation. Besides the competitive results on public benchmarks, SEED-X demonstrates its effectiveness in handling real-world applications across various domains. We hope that our work will inspire future research into what can be achieved by versatile multimodal foundation models in real-world applications. All models, training, and inference codes are available at `https://anonymous.4open.science/r/SEED-X/`.

## 1 INTRODUCTION

In recent years, Multimodal Large Language Models (MLLMs) (Li et al., 2023e; Zhu et al., 2023a; Liu et al., 2023b; Peng et al., 2023; Bai et al., 2023; Liu et al., 2023a; Zhang et al., 2023b; Lin et al., 2023) have demonstrated exceptional capabilities in comprehending multimodal data through leveraging the strong generality of LLMs (Touvron et al., 2023; Brown et al., 2020; Chowdhery et al., 2022). Some pioneering work (Sun et al., 2023b; Yu et al., 2023a; Ge et al., 2023a;b; Wu et al., 2023; Dong et al., 2023; Sun et al., 2023c; Zhu et al., 2023b) further empower LLMs with the ability to generate images beyond texts. While these models can handle a variety of tasks and excel in academic benchmarks, the accuracy and diversity of their generated content still fall short of real-world needs. We argue that further research on versatile multimodal foundation models should focus more on bridging this gap.

**What characteristics should a multimodal foundation model possess to be applicable in real-world scenarios?** We posit that it should tackle the inherent challenge of capturing multi-granularity visual semantics for both comprehension and generation tasks, given that a multimodal foundation model has to accommodate various downstream tasks requiring different levels of visual semantics. As a result, two essential features should be incorporated into the model design: (1) understanding images of arbitrary sizes and ratios, and (2) multi-granularity image generation, encompassing both high-level instructional image generation and low-level image manipulation tasks. These attributes form the basis for a multimodal foundation model's effective application in an open-world context.

In this paper, we introduce SEED-X, a unified and versatile multimodal foundation model that seamlessly integrates the essential features mentioned above. Specifically, SEED-X supports object detection and dynamic resolution image encoding for multi-granularity comprehension, as well as high-level instructional image generation and low-level image manipulation for multi-granularity image generation. It is important to emphasize that *integrating all these characteristics into a single foundation model is by no means trivial*, as highlighted in Table 1, since none of the previous works fully support all of these features.

After instruction tuning, SEED-X can function as multimodal AI assistants in the real world, capable of addressing various user needs through generating proper texts and images as shown in Fig. 1. Specifically, our instruction-tuned models can act as an interactive designer, generating images while illustrating creative intent, offering modification suggestions and showcasing visualizations based on user's input images. Additionally, they can act as knowledgeable personal assistants, comprehending images of various sizes and providing relevant suggestions. Moreover, they can generate more diverse outputs, such as slide layouts for slide creation, and interleaved image-text content for storytelling. SEED-X signifies a notable advancement in developing a versatile agent for users in the real world.

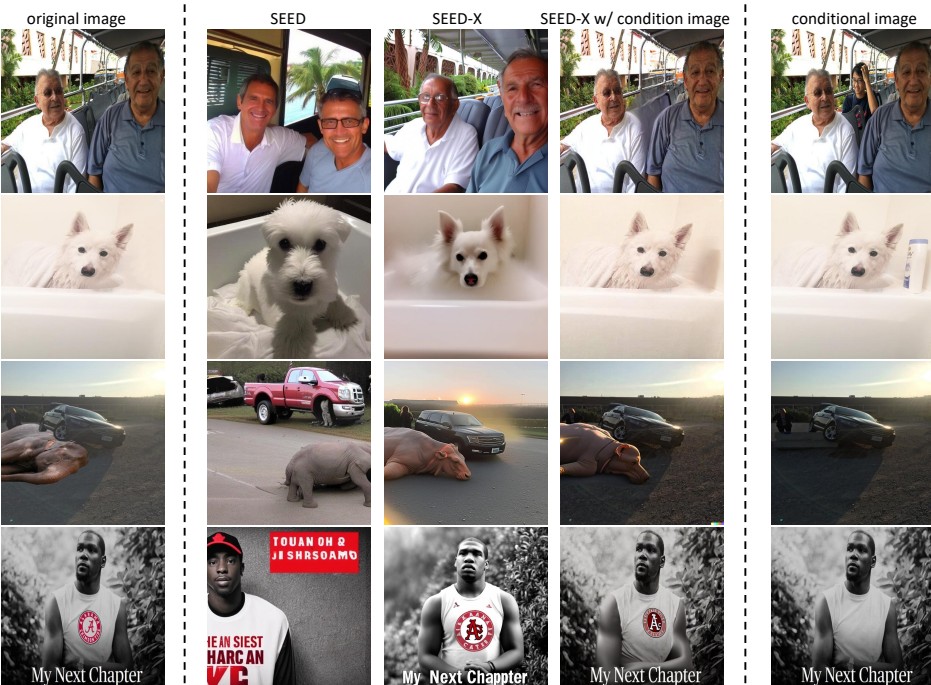

Figure 2: The reconstruction results of our visual de-tokenizer. It can decode realistic images that are semantically aligned with the original images by taking the **ViT features** as inputs, and further recover fine-grained details by incorporating the **conditional images** as inputs.

To endow SEED-X with the aforementioned characteristics, our approach incorporates (1) a visual tokenizer to unify image comprehension and generation, where its multi-granularity de-tokenization phase facilitates image generation and high-precision image manipulation, and (2) an MLLM with dynamic resolution image encoding to enable the comprehension of images with arbitrary sizes and aspect ratios. Specifically, we utilize a pre-trained ViT as the visual tokenizer and train a visual de-tokenizer to decode realistic images by taking the ViT features as input. To realize the retention of fine-grained details of the input image to satisfy image manipulation, we further fine-tune the visual de-tokenizer to take an extra condition image as input in the latent space (See Fig. 2). The ViT features **serve as a bridge** to decouple the training of the visual (de-)tokenizer and the MLLM. The dynamic resolution image encoding divides an input image into sub-images and adds extrapolatable 2D positional embeddings to the ViT features of each sub-image, allowing the MLLM to scale to any image resolution. For image generation, a fixed number of learnable queries are fed into the MLLM, where the output hidden states are trained to reconstruct the ViT features of the target images.

We pre-train SEED-X on massive multimodal data, including image-caption pairs, grounded image-text data, interleaved image-text data, OCR data, and pure texts. We further apply multimodal instruction tuning to align SEED-X with human instructions across various domains, utilizing both existing datasets and newly collected datasets that cover image editing, text-rich, grounded and referencing QA, and slide generation tasks. The extensive evaluations on MLLM benchmarks demonstrate that our instruction-tuned model not only achieves competitive performance in multimodal comprehension, but also achieves state-of-the-art results in image generation compared to existing MLLMs on SEED-Bench-2 (Li et al., 2023c).

All models, training and inference codes are available at `https://anonymous.4open.science/r/SEED-X/`. We hope our work can bring insights about the potential of multimodal models in real-world scenarios through unifying multi-granularity comprehension and generation.

## 2 RELATED WORK

With the rapid development of Multimodal Large Language Models (MLLM), recent studies have been working on unified MLLMs that are capable of **multimodal comprehension and generation**

Table 1: MLLMs that unify comprehension and generation listed by publication date and whether they support the significant characteristics essential for real-world applications. "Decoder Input" denotes the inputs for image generation, where "Features" means continuous features, "Token" represents discrete tokens, "Text" implies text prompts, and "Latent" denotes VAE latent. "-" indicates that we are unsure whether the model supports this characteristic.

| | Date | Decoder Input | Detection | Dynamic-Res Img Input | Image Gen | High-precision Editing | Open-source |
|---|---|---|---|---|---|---|---|
| Emu | 07/2023 | Feature | ✗ | ✗ | ✓ | ✗ | ✓ |
| CM3Leon | 07/ 2023 | Token | ✗ | ✗ | ✓ | ✗ | ✗ |
| SEED-OPT | 07/ 2023 | Token | ✗ | ✗ | ✓ | ✗ | ✗ |
| LaVIT | 09/2023 | Token | ✗ | ✗ | ✓ | ✗ | ✓ |
| NExT-GPT | 09/2023 | Feature | ✗ | ✗ | ✓ | ✗ | ✓ |
| DreamLLM | 09/2023 | Feature | ✗ | ✗ | ✓ | ✗ | ✗ |
| SEED-LLaMA | 10/2023 | Token | ✗ | ✗ | ✓ | ✗ | ✓ |
| VL-GPT | 12/2023 | Feature | ✗ | ✗ | ✓ | ✗ | ✗ |
| Gemini | 12/2023 | Token | ✗ | - | ✓ | ✗ | ✗ |
| Emu2 | 12/2023 | Feature | ✗ | ✗ | ✓ | ✗ | ✓ |
| Unified-IO 2 | 12/2023 | Token | ✓ | ✗ | ✓ | ✗ | ✓ |
| Mini-Gemini | 03/2024 | Text | ✗ | ✗ | ✓ | ✗ | ✓ |
| Chameleon | 05/ 2023 | Token | ✗ | ✗ | ✓ | ✗ | ✓ |
| Transfusion | 08/2024 | Latent | ✗ | ✗ | ✓ | ✓ | ✗ |
| Show-o | 08/2024 | Token | ✗ | ✗ | ✓ | ✗ | ✓ |
| VILA-U | 09/2024 | Token | ✗ | ✗ | ✓ | ✗ | ✗ |
| **SEED-X** | 09/2024 | Feature | ✓ | ✓ | ✓ | ✓ | ✓ |

as shown in Tab. 1. Some work (Ge et al., 2023b;a; Yu et al., 2023a; Jin et al., 2023; Lu et al., 2023; Team, 2024; Xie et al., 2024; Wu et al., 2024) utilize a discrete visual tokenizer to perform multimodal autoregression with a unified next-word-prediction objective or masked visual token prediction. Some research efforts (Sun et al., 2023b;a; Zhu et al., 2023b) have delved into multimodal autoregression with continuous representations, where each image in the multimodal sequence is tokenized into embeddings via a visual encoder, and then interleaved with text tokens for autoregressive modeling. During inference, the regressed visual embeddings will be decoded into an image by a visual decoder. Additionally, some studies (Dong et al., 2023; Wu et al., 2023) enable image generation in a non-autoregressive manner through utilizing learnable queries to obtain visual representations from MLLMs, which are further fed into a image decoder to generate images. Mini-Gemini, generates text prompts using MLLMs and then leverages the existing SDXL (Podell et al., 2023) to output images. Recent work Transfusion Zhou et al. (2024) adopts diffusion objectives, where the noised image latents are de-noised for image generation through a VAE decoder.

Although these work have achieved competitive results on various academic benchmarks, such as VQA and text-to-image generation, the accuracy and diversity of their generated content still fall short of real-world needs, since they do not meet the requirements of modeling multi-granularity visual semantics for comprehension and generation task. As shown in Tab. 1, we identify several significant characteristics essential for real-world applications including object detection and dynamic resolution image encoding for multi-granularity comprehension, as well as high-level instructional image generation and low-level image manipulation for multi-granularity image generation. Notably, **none of the previous works fully support all of these characteristics**. In this work, we present SEED-X, a unified and versatile foundation model, which effectively incorporate the aforementioned characteristics for real-world applications.

## 3 METHOD

### 3.1 VISUAL TOKENIZATION AND DE-TOKENIZATION

In SEED-X, we adopt a visual tokenizer to unify image comprehension and generation, and pre-train a multi-granularity de-tokenizer to facilitate image generation and high-precision image manipulation

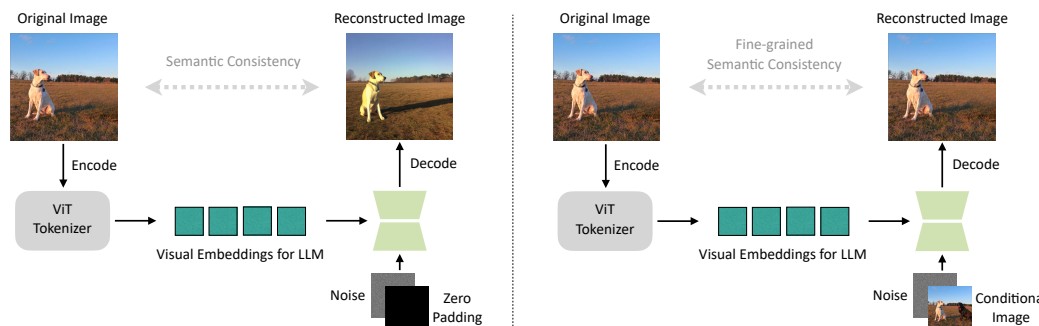

Figure 3: Overview of visual tokenization and de-tokenization in SEED-X. In the first stage (left), we pre-train a visual de-tokenizer, which can decode semantically consistent images by taking the features of a pre-trained ViT as inputs. In the second stage (right), we fine-tune the visual de-tokenizer through concatenating the latent features of a conditional image with the noise to recover the fine-grained details of the original image.

in a two-stage manner. In the first stage, as shown in Fig. 3 (left), we utilize a pre-trained ViT as the visual tokenizer and pre-train a visual de-tokenizer to decode realistic images by taking the features of the ViT as inputs in the first stage. Specifically, $N$ visual embeddings from the ViT tokenizer ($N = 64$ after average pooling) are fed into a learnable module as the inputs of the U-Net of the pre-trained SD-XL (Podell et al., 2023) (replacing the original text features). The learnable module consists of four cross-attention layers to connect the visual tokenizer and the U-Net. We optimize the parameters of the learnable module and keys and values within the U-Net on the images from JourneyDB (Sun et al., 2024), LAION-Aesthetics (Schuhmann & Beaumont, 2022), Unsplash (Ali et al., 2023), and LAION-COCO (Schuhmann et al., 2023). As shown in Fig. 2, compared with SEED (Ge et al., 2023b), our visual de-tokenizer can decode images that are more semantically aligned with the original images by taking the ViT features as inputs.

In the second stage, as shown in Fig. 3 (right), we further fine-tune the visual de-tokenizer to take an extra condition image as inputs for the retention of low-level details. Specifically, we follow InstructPix2Pix (Brooks et al., 2023) to encode the condition image into the latent space via the VAE encoder, and concatenate them with the noisy latent as the input of U-Net. The channel number of the U-Net convolutional layer is expanded from 4 to 8, and all parameters of U-Net are optimized. We fine-tune the visual de-tokenizer on MagicBrush (Zhang et al., 2023a) and in-house image editing data, as well as the pure images in the first stage, where the conditional inputs are set to zeros. As shown in Fig. 2, by incorporating the condition image as an additional input besides the high-level image features, our visual de-tokenizer can recover the fine-grained details of the original image.

## 3.2 DYNAMIC RESOLUTION IMAGE ENCODING

Current MLLMs require to resize the input images to a pre-defined resolution (typically a square size), which corresponds to the training resolution of the vision encoder, which can result in the loss of fine-grained information. In this work, we propose dynamic resolution image encoding to enable the processing of images with arbitrary sizes and aspect ratios by dividing the image into a grid comprising of sub-images. Specifically, for the visual encoder with the training resolution $H_t \times W_t$, we first up-sample the input image with the size $H \times W$ to the size of $\{N_h * H_t\} \times \{N_w * W_t\}$. The grid size $N_h \times N_w$, are determined by

$$\min \quad N_h * N_w,$$
$$\text{s.t. } H \le N_h * H_t \quad \text{and} \quad W \le N_w * W_t. \tag{1}$$

We also resize the original image to the size of $H_t \times W_t$ to provide global visual context. All sub-images and the resized global image are fed into the visual encoder to obtain the features, which are concatenated as the input of the LLM.

To enable the LLM to be aware of the positional information of each sub-image within the original image, we add extrapolatable 2D positional embeddings to the visual features of each sub-image. Specifically, for a sub-image with a normalized center location $(x_c, y_c)$ in the grid, where $0.0 <$

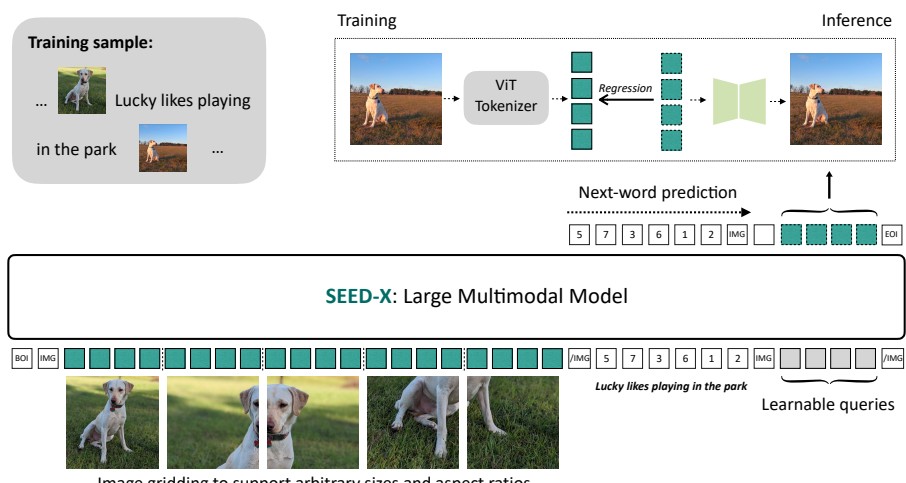

Figure 4: Overview of SEED-X for multimodal pre-training. Each image is divided into sub-images to support arbitrary sizes and aspect ratios, and their ViT features along with text tokens are fed into an LLM to perform next-word prediction and image feature regression between the output hidden states of the learnable queries and ViT features. During inference, the regressed image features are fed into the visual de-tokenizer to decode images.

$x_c, y_c < 1.0$, its learnable positional embedding $p$ is computed:

$$p = x_c * l + (1 - x_c) * r + y_c * t + (1 - y_c) * b. \tag{2}$$

$l$, $r$, $t$, and $b$ represent four learnable position embeddings indicating left, right, top and bottom respectively. Consequently, our visual encoder can handle inputs with any arbitrary sizes and aspect ratios, even if the image resolution was not encountered during training.

### 3.3 MULTIMODAL PRE-TRAINING AND INSTRUCTION TUNING

#### 3.3.1 TRAINING STAGE I: MULTIMODAL PRE-TRAINING

As shown in Fig. 4, SEED-X adopts next-word prediction and image feature regression training objectives on interleaved visual and textual data. Specifically, we perform dynamic resolution encoding of each image in the multimodal sequence, and their features along with text tokens are fed into the pretrained LLM. In order to equip the model with detection and referencing abilities, we add 224 bbox tokens, designated for representing bounding box coordinates, represented by <box_start> <loc-x_center> <loc-y_center> <loc-width> <loc-height> <box_end> with special tokens at the beginning and end of the bounding box. The text and added bbox tokens are trained through predicting the next token with cross-entropy loss.

We employ $N$ learnable queries ($N = 64$ to align with the visual de-tokenizer) to obtain the output visual representations from the LLM, which are trained to reconstruct the features of the pre-trained ViT tokenizer with a Mean Squared Error (MSE) loss. We add two special tokens '' and '</IMG>' to represent the beginning and the end of the query embeddings, and the '' is trained to predict where an image emerges. In doing so, we utilize the pre-trained ViT tokenizer as a **bridge** to decouple the training of a visual de-tokenizer and the MLLM for image generation. During inference, the regressed visual representations from SEED-X are fed into the visual de-tokenizer to decode realistic images.

We pre-train SEED-X initialized from Llama2-chat-13B using LoRA on massive multimodal data, including image-captions pairs, grounded image-texts, interleaved image-text data, OCR data and pure texts. We perform pre-training with 128 A100-40G GPUs (4 days) on a total of 120M samples. See Appendix. A.1 and Appendix. A.2 for more details.

Table 2: Comparison for multimodal comprehension and generation on MLLM benchmarks. "Image Gen" denotes whether the model can generate images besides texts. "Single", "Multi" and "Interleaved" denote evaluating the comprehension of single-image, multi-image, and interleaved image-text. "Gen" denotes evaluating the generation of images, and "/" denotes the model's inability to perform such evaluation. The best results are **bold** and the second best results are underlined.

| | Size | Image Gen | MMB | SEED-Bench-2 | | | | MME | |
|---|---|---|---|---|---|---|---|---|---|
| | | | | P1 | | P2 | P3 | Perce-ption | Cog-nition |
| | | | Single | Single | Multi | Inter-leaved | Gen | Single | Single |
| GPT-4v | - | × | **77.0** | **69.8** | **73.1** | 37.9 | / | 1409 | 517 |
| Gemini Pro | - | ✓ | 73.6 | 62.5 | - | - | - | **1609** | **540** |
| Qwen-VL-Chat | 10B | × | 61.8 | 50.3 | 37.4 | 38.5 | / | 1488 | 361 |
| Next-GPT | 13B | ✓ | - | 31.0 | 27.8 | 40.3 | 42.8 | - | - |
| Emu | 14B | ✓ | - | 46.4 | 31.2 | 45.6 | 45.7 | - | - |
| SEED-LLaMA-I | 14B | ✓ | - | 49.9 | 32.4 | **48.0** | 50.6 | - | - |
| LLaVA-1.5 | 8B | × | 66.5 | 58.3 | 39.2 | 34.4 | / | 1506 | 302 |
| XComposer-VL | 8B | × | 74.4 | 66.5 | 50.0 | 29.0 | / | 1528 | 391 |
| SPHINX-1k | - | × | 67.1 | 68.5 | 37.7 | 32.5 | / | 1560 | 310 |
| Emu2-Chat | 37B | × | 63.6 | - | - | - | / | 1345 | 333 |
| SEED-X | 17B | ✓ | 65.8 | 48.2 | 53.8 | 24.3 | 57.8 | 1250 | 236 |
| SEED-X-I | 17B | ✓ | 70.1 | 64.2 | 57.3 | 39.8 | **62.8** | 1457 | 321 |

### 3.3.2 TRAINING STAGE II: MULTIMODAL INSTRUCTION TUNING

We perform multimodal instruction tuning through fine-tuning SEED-X using a LoRA module with both public datasets and in-house data covering image editing, text-rich, grounded and referencing QA, and slide generation tasks. The details of datasets can be found in Appendix. A.1. We fine-tune SEED-X with conversational and image generation data to yield a general instruction-tuned model SEED-X-I, which can follow multimodal instructions and make responses with images, texts and bounding boxes in multi-turn conversation. We further fine-tune the foundation model SEED-X on specialized datasets, resulting in a series of instruction-tuned models tailored for specific tasks, including SEED-X-Edit, SEED-X-PPT, SEED-X-Story and SEED-X-Try-on. The proficient capabilities of these instruction-tuned model across various domains demonstrate the versatility of our pre-trained foundation model SEED-X.

## 4 EXPERIMENTS

### 4.1 QUANTITATIVE EVALUATION

We evaluate SEED-X-I on benchmarks specifically designed for evaluating MLLMs, since recent work (Liu et al., 2023c; Li et al., 2023c) point out that traditional VQA benchmarks are not tailored for evaluating MLLMs with open-form output. As shown in Tab. 2, SEED-X-I achieves competitive performance in both the image comprehension and generation tasks. For example, it achieves an accuracy rate of over 70% on MMBench (Liu et al., 2023c) for evaluating single-image understanding. SEED-X-I also shows promising results for comprehending multi-image and interleaved image-text content in SEED-Bench-2 (Li et al., 2023b). Compared with previous work (Sun et al., 2023c; Ge et al., 2023b; Wu et al., 2023) that unify comprehension and generation within an LLM, SEED-X-I achieves the state-of-the-art performance in P3 level of SEED-Bench-2 including the evaluation of text-to-image generation, next image prediction and text-image creation.

### 4.2 QUALITATIVE EVALUATION

#### 4.2.1 APPLICATIONS IN THE REAL WORLD.

Since SEED-X seamlessly integrates two essential features including the comprehension of images of arbitrary sizes and ratios, and multi-granularity image generation, encompassing both high-

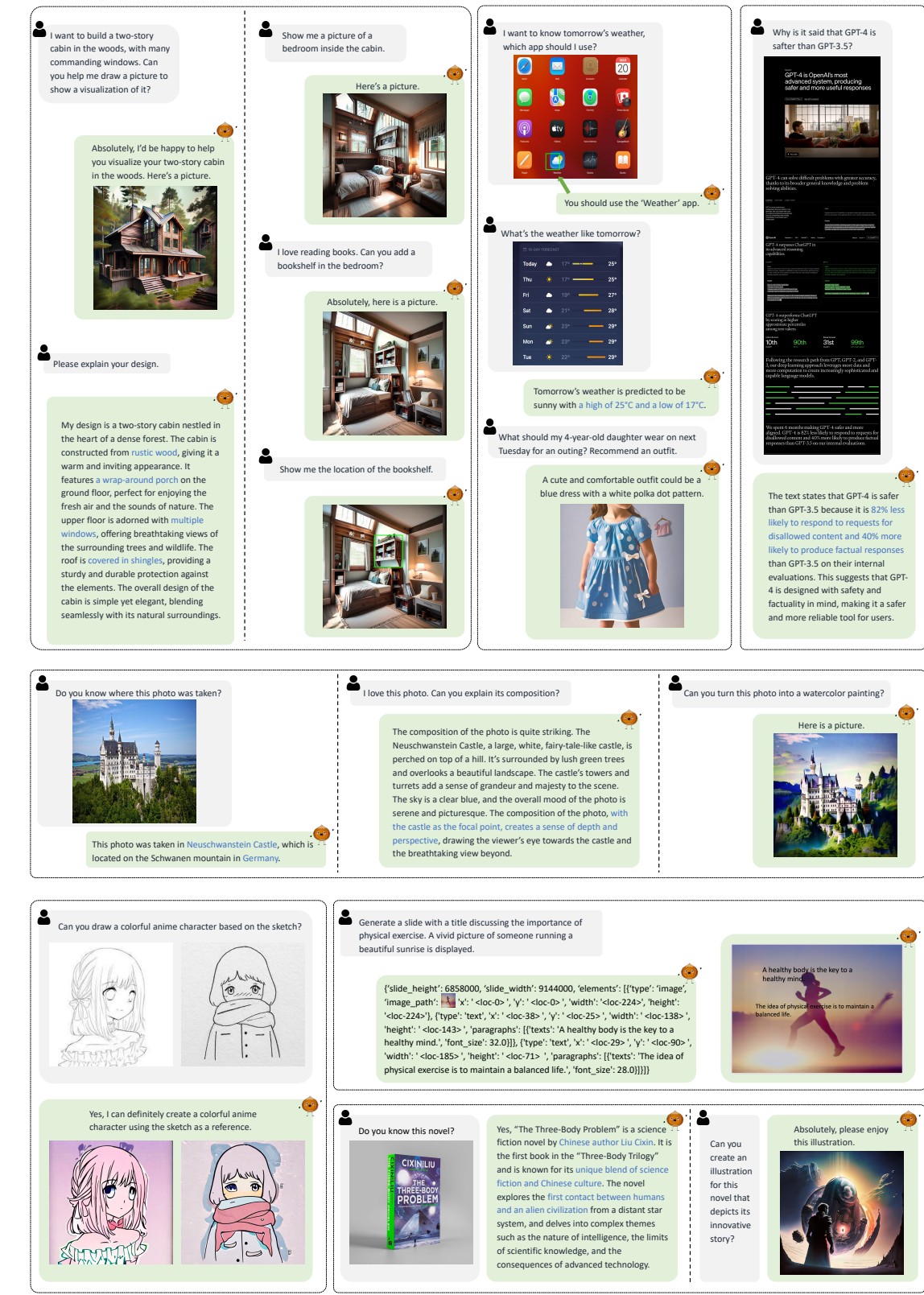

Figure 5: Examples of what SEED-X can do in real-world scenarios after instruction tuning through unifying multi-granularity comprehension and generation. Our instruction tuned models can function as an interactive designer, generating images without descriptive captions while illustrating creative intent, and showcasing visualizations of modified images. They can act as knowledgeable personal assistants, comprehending images of arbitrary sizes and offering relevant suggestions in multi-turn conversations.

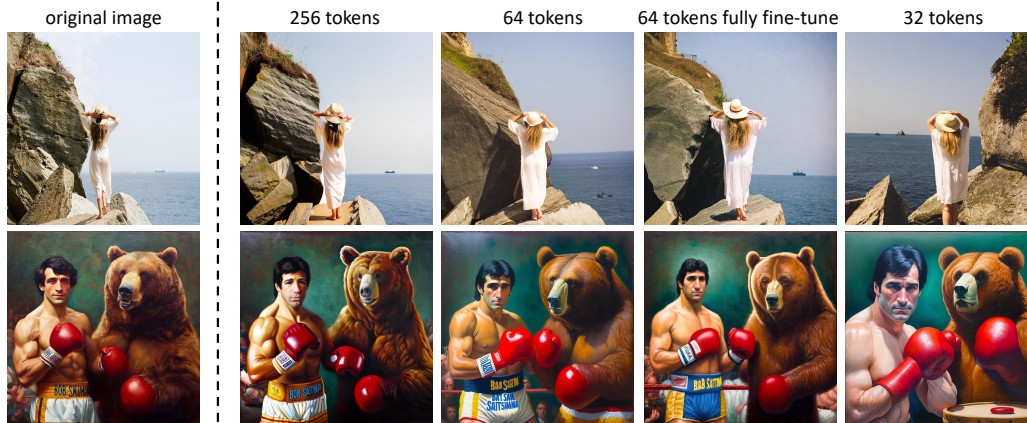

Figure 6: Ablation study on the number of visual tokens and trainable parameters for training visual de-tokenizer.

level instructional image generation and low-level image manipulation tasks, it can be effectively instruction tuned to function as multimodal AI assistants in the real world across various domains. As shown in Fig. 1 and Fig. 5, our instruction tuned models can serve as an interactive designer, which can generate images without descriptive captions while illustrate creative intent, and showcase visualizations of modified images. For example, it can explain the design idea of concept image for AGI and a two-story cabin. It can create an imaginative illustration for the novel without the need of describing the scene with languages. It can further offer modification suggestions of the user's room and showcase the visualization. Additionally, the instruction tuned models can act as an knowledgeable personal assistant, comprehending images of arbitrary sizes and providing relevant suggestions. For example, it can identify foods suitable for fat reduction in the refrigerator, display appropriate clothing based on the screenshot of weather forecasts.

### 4.2.2 IMAGE GENERATION AND MANIPULATION.

We compare previous MLLMs that are capable of generating images for text-to-image generation in Fig. 8 of Appendix. Our instruction tuned model can generate images that are more aligned with the elements in the caption and possess artistic qualities. Through utilizing a pre-trained ViT Tokenizer as the bridge to decouple the training of visual de-tokenizer and the MLLM, our pre-trained model SEED-X can effectively realize high-quality image generation, which is a fundamental capability to be applied in real-world scenarios.

We further compare image manipulation with previous MLLMs (See Appendix. A.3). As shown in Fig. 9, we can observe that SEED-X-Edit can more effectively adhere to editing instructions while maintaining the low-level details of the input image. Our MLLM accurately predicts visual semantic representations based on an input image and a language instruction, which serve as input for the U-Net. The visual de-tokenizer can further condition on the input image, ensuring the preservation of fine-grained details in the decoded images.

### 4.2.3 MULTIMODAL COMPREHENSION.

We provide qualitative examples of multimodal comprehension by SEED-X-I in Fig. 10 and Fig. 11 of Appendix. SEED-X-I can realize fine-grained object detection and perception, text-rich comprehension, fundamental mathematical computation, world-knowledge and commonsense reasoning, diagram understanding, etc.

### 4.3 ABLATION STUDY

In this section, we perform ablation studies on the training of our visual de-tokenizer and the pre-training of SEED-X to enable a MLLM for image generation.

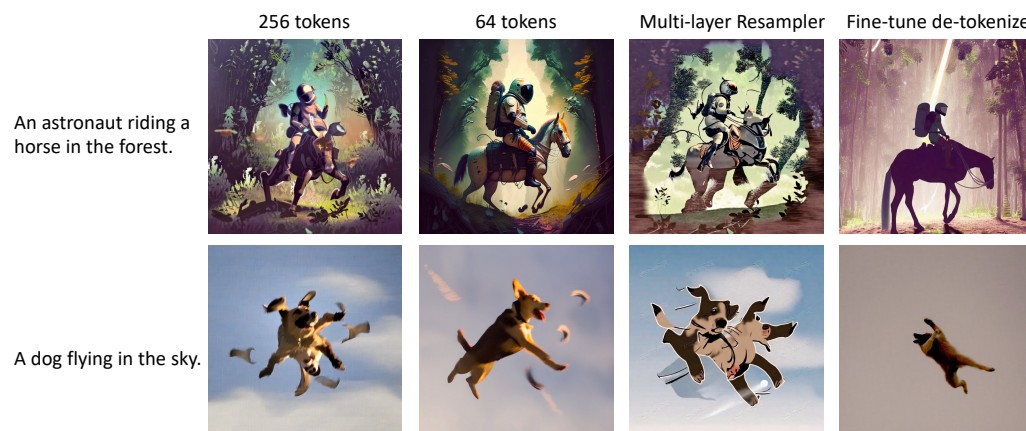

Figure 7: Ablation study on the number of visual tokens, model architecture and optimization targets during pre-training SEED-X for image generation.

For visual de-tokenization, N visual embeddings (after average pooling) from the ViT tokenizer are fed into a learnable module as the inputs of the U-Net of the pre-trained SD-XL. We perform an ablation study on the number of visual tokens and the learnable parameters of the SD-XL U-Net, where keys and values within the U-Net are optimized if not specified with "fully fine-tune". As shown in Fig. 6, we can observe that more visual tokens can result in better reconstruction of the original images. For example, the decoded images from 256 visual embeddings can recover the characters' postures of the original images, while decoded images from 32 visual embeddings have already lost the original structure of the scene. We further observe that fully fine-tuning the parameters of the SD-XL U-Net can lead to distortions in image details, such as the woman's feet, compared to only training the keys and values within the U-Net. In SEED-X, we use N = 64 visual embeddings to train the visual de-tokenizer and only optimize the keys and values within the U-Net (See below for an explanation of why we do not choose N = 256).

To enable MLLM for image generation, we employ N learnable queries to obtain the output visual representations from the LLM, which are trained to reconstruct N visual embeddings from the ViT tokenizer with a learnable module. We first perform an ablation study on the number of learnable queries. The images generated by the MLLM based on the input caption are shown in Fig. 7. We can observe that using 256 learnable queries to reconstruct 256 visual embeddings can lead to distortion in the generated images compared with N = 64. This occurs because regressing more visual features is more challenging for the model, even though 256 visual embeddings from the de-tokenizer can better reconstruct images, as demonstrated in the previous ablation study. We also observe that, compared to learning a one-layer cross-attention for reconstructing image features, a multi-layer resampler (multi-layer cross-attention) yields less satisfactory performance, which can happen due to the lack of more direct regularizations on the hidden states of the LLM. We further optimize the visual de-tokenizer by using the reconstructed visual embeddings from the MLLM as input instead of ViT features, but the generated images exhibit a more monotonous appearance. It demonstrates the effectiveness of utilizing the ViT Tokenizer as the bridge to decouple the training of visual de-tokenizer and the MLLM for image generation.

## 5 CONCLUSION

In this paper, we present SEED-X, a versatile foundation model, which can function as multimodal AI assistants in the real world after instruction tuning. SEED-X seamlessly integrates two essential features including image comprehension of arbitrary sizes and ratios, and multi-granularity image generation, which encompasses both high-level instructional image generation and low-level image manipulation tasks. These fundamental features form the basis for a multimodal foundation model to be effectively applied in an open-world context. We hope that SEED-X can inspire future research into the potential of multimodal large language models (MLLMs) in the real-world scenarios through unifying multi-granularity comprehension and generation.

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

# A APPENDIX

## A.1 PRE-TRAINING AND INSTRUCTION TUNING DATASETS

As listed in Tab. 3, we pre-train SEED-X and conduct instruction tuning on a large variety of both public datasets and in-house data. For multimodal pre-training, we utilize image-caption pairs, grounded image-caption pairs, interleaved image and text content, OCR data and pure text data. The images of LAION-COCO (Christoph et al., 2022) and SAM (Kirillov et al., 2023) are re-captioned with Qwen-VL-Chat (Bai et al., 2023) for a more detailed descriptive caption to improve both image comprehension and generation.

For instruction tuning, we utilize various public VQA datasets, and further curate text-rich QA, grounded and referencing QA to enhance the model's capability of comprehending text-rich images and detecting objects that requires reasoning. We use multiple conversational datasets, which are specifically collected for MLLMs with open-form text output. We use the same image-caption pairs as in the pre-training phase to maintain the model's ability to generate images. For the image manipulation, since the high-precision editing dataset MagicBrush (Zhang et al., 2023a) is only at the level of thousands, we employ a series of models to collect a dataset of millions of image editing examples, which are used for both training the visual de-tokenizer and SEED-X-Edit. We further collected data on slides, obtaining images, captions, and layouts for training slide generation.

## A.2 IMPLEMENTATION DETAILS

**Visual Tokenization and De-tokenization.** We use the visual encoder from Qwen-vl (Bai et al., 2023) as the ViT Tokenizer and adopt 1D average pooling to obtain $N = 64$ visual embeddings. These visual embeddings are fed into four layers of cross-attention as the input of the U-Net initialized from SDXL (Podell et al., 2023). In the first stage, we optimize the parameters of the cross-attention layers and the keys and values within the U-Net on the images from JourneyDB (Sun et al., 2024), LAION-Aesthetics (Schuhmann & Beaumont, 2022), Unsplash (Ali et al., 2023), and LAION-COCO (Schuhmann et al., 2023). We train the visual de-tokenizer on 32 A100-40G GPUs with 42K training steps, where the learning rate is set to 1e-4 with cosine decay.

In the second stage, we encode the condition image into the latent space via the VAE encoder, and concatenate them with the noisy latent as the input of U-Net. The channel number of the U-Net convolutional layer is expanded from 4 to 8, and all parameters of U-Net are optimized. We pre-train the visual conditioner on MagicBrush (Zhang et al., 2023a) and in-house image editing data, as well as the image-caption pairs in the first stage, where the conditional inputs are set to zeros. We fine-tune the visual de-tokenizer on 32 A100-40G GPUs with 30K training steps, where the learning rate is set to 1e-4 with cosine decay.

**Multimodal Pre-training and Instruction Tuning.** We utilize the visual encoder from Qwen-vl (Bai et al., 2023) as the ViT Tokenizer and initialize a cross-attention layer to obtain $N = 64$ visual embedding as the input of the LLM initialized from Llama2-chat-13B. We initialize $N = 64$ learnable queries and the output hidden states from them are fed into a cross-attention layer to reconstruct $N = 64$ visual embeddings from the ViT Tokenizer. We optimize the LLM using LoRA and optimize the parameters of the input cross-attention layer, output cross-attention layer, extrapolatable 2D positional embeddings, and LoRA on image-captions pairs, grounded image-texts, interleaved image-text data, OCR data and pure texts. We perform pre-training with 128 A100-40G GPUs (4 days) on a total of 120M samples, where the learning rate is set to 1e-4 with cosine decay.

For the instruction tuning, we fine-tune a LoRA module on the pre-trained model, and optimize the parameters of the input cross-attention layer, output cross-attention layer, extrapolatable 2D positional embeddings, and LoRA. We further fine-tune SEED-X on specialized datasets, resulting in a series of instruction-tuned models tailored for specific tasks, including SEED-X-Edit, SEED-X-PPT, SEED-X-Story and SEED-X-Try-on.

## A.3 QUALITATIVE EXAMPLES

**Text-to-image Generation,** Fig. 8 visualizes the comparison between MLLMs for text-to-image generation including Next-GPT (Wu et al., 2023), SEED-LLaMA-I(Ge et al., 2023b), Emu2-Gen (Sun

Table 3: Overview of the pre-training and instruction tuning datasets.

| Type | Dataset |
|---|---|
| **Pre-training** | |
| Image-Caption | LAION-COCO (Christoph et al., 2022) (Re-caption), SAM (Kirillov et al., 2023) (Re-caption), LAION-Aesthetics(Schuhmann & Beaumont, 2022), Unsplash (Ali et al., 2023), JourneyDB (Pan et al., 2023), CapFusion (Yu et al., 2023b), |
| Grounded Image-Caption | GRIT (Peng et al., 2023) |
| Interleaved Image-Text | MMC4 (Zhu et al., 2023c), OBELICS (Laurençon et al., 2023), OpenFlamingo (Awadalla et al., 2023) |
| OCR | LLaVAR (Zhang et al., 2023c), Slides (In-house) |
| Pure Text | Wikipedi |
| **Instruction Tuning** | |
| VQA | LLaVAR (Zhang et al., 2023c), Text-rich QA (In-house), MIMIC-IT (Li et al., 2023a), MathQA (Amini et al., 2019), ChartQA (Masry et al., 2022), AI2D (Kembhavi et al., 2016), ScienceQA (Lu et al., 2022), KVQA (Shah et al., 2019), DVQA (Kafle et al., 2018), Grounded QA (In-house) Referencing QA (In-house) |
| Conversation | LLaVA-150k (Liu et al., 2024), ShareGPT (Chen et al., 2023), VLIT (Li et al., 2023d), LVIS-Instruct4V (Wang et al., 2023), Vision-Flan (Xu et al., 2024), ALLaVA-4V (Chen et al., 2024) |
| Image Generation | LAION-COCO (Christoph et al., 2022) (Re-caption), SAM (Kirillov et al., 2023) (Re-caption), LAION-Aesthetics(Schuhmann & Beaumont, 2022), Unsplash (Ali et al., 2023), JourneyDB (Pan et al., 2023) |
| Image Editing | Instructpix2pix (Brooks et al., 2023), MagicBrush (Zhang et al., 2023a), Openimages (Kuznetsova et al., 2020)-editing (In-house), Unsplash (Ali et al., 2023)-editing (In-house) |
| Slides Generation | In-house data |
| Story Telling | VIST (Huang et al., 2016) |
| Virtual Try-on | VITON-HD (Choi et al., 2021) |

et al., 2023a) and Gemini (Team et al., 2023). Compared with previous MLLMs, our instruction tuned model can generate images that are more aligned with the elements in the descriptive caption and possess artistic qualities. For example, images generated by SEED-X-I vividly and accurately depicts "person standing in a small boat", "a gleaming sword on its back", "an oriental landscape painting", "tiger with vivid colors" in the captions. Through utilizing a pre-trained ViT Tokenizer as the bridge to decouple the training of visual de-tokenizer and the MLLM, our pre-trained model SEED-X can effectively realize high-quality image generation, which is a fundamental capability for applying multimodal models in real-world scenarios.

| Next-GPT | SEED-LLaMA-I | Emu2-Gen | Gemini | SEED-X-I |
|---|---|---|---|---|

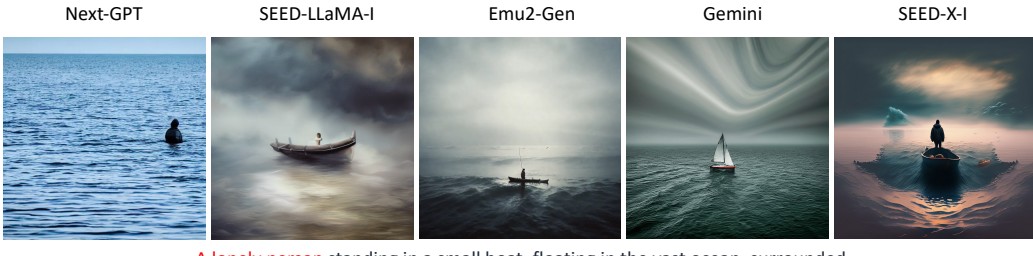

A lonely person standing in a small boat, floating in the vast ocean, surrounded by thick fog in the sky, giving a sense of confusion and helplessness.

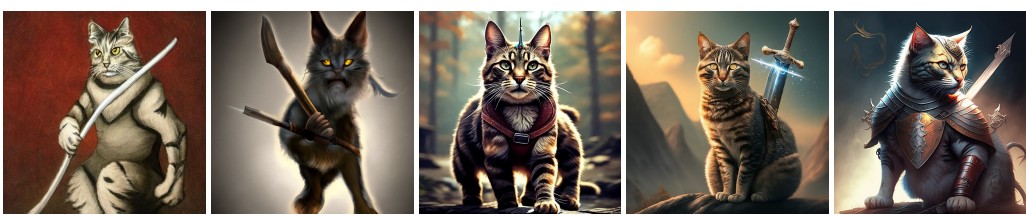

A fearless cat, with a gleaming sword on its back.

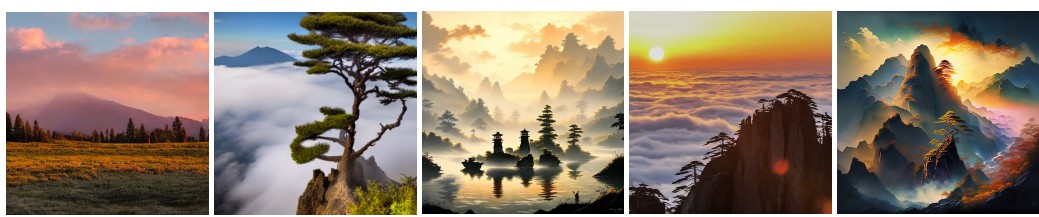

The golden moment of sunrise, Huangshan, China, towering Qishi Peak, a large area of clouds, a small welcome pine, an oriental landscape painting.

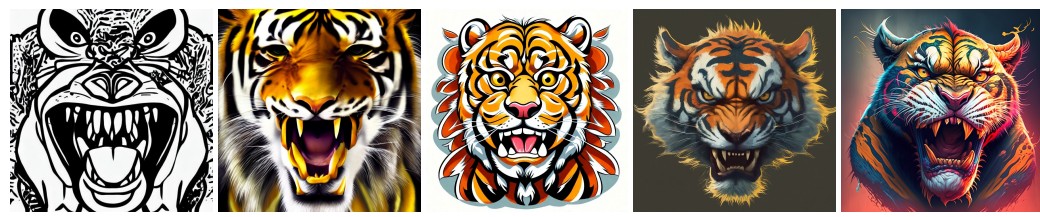

A fierce cartoon tiger, with furrowed brows and bared teeth, displays its anger through vivid colors and exaggerated features.

Figure 8: Qualitative comparison between MLLMs for text-to-image generation. SEED-X-I is capable of generating images that are more closely aligned with the elements in the descriptive caption and exhibit artistic qualities.

**Image manipulation.** We compare image manipulation with previous MLLMs including Emu2-Gen (Sun et al., 2023a), Gemini (Team et al., 2023), MGIE (Team et al., 2023) and Mini-Gemini (Team et al., 2023). Language-guided image manipulation presents a significant challenge as the model must be capable of comprehending free-form instructions and generating images with the low-level details of the input image preserved. As shown in Fig. 9, we can observe that SEED-X-Edit can more effectively adhere to editing instructions while maintaining the low-level details of the input image. For instance, SEED-X-Edit can accurately add sunglasses to the dog on the right, while both Emu2-Gen and MGIE fail to follow the instruction, resulting in sunglasses being added to both dogs. Additionally, SEED-X-Edit successfully eliminates the dog in the baby image while preserving the low-level background details and the baby's features. In contrast, Emu2-Gen fails to retain the fine details of the input image, and MGIE is unsuccessful in removing the dog. Note that Gemini lacks the ability to edit images as it **retrieves images** on the Internet. Here the presence of black images is

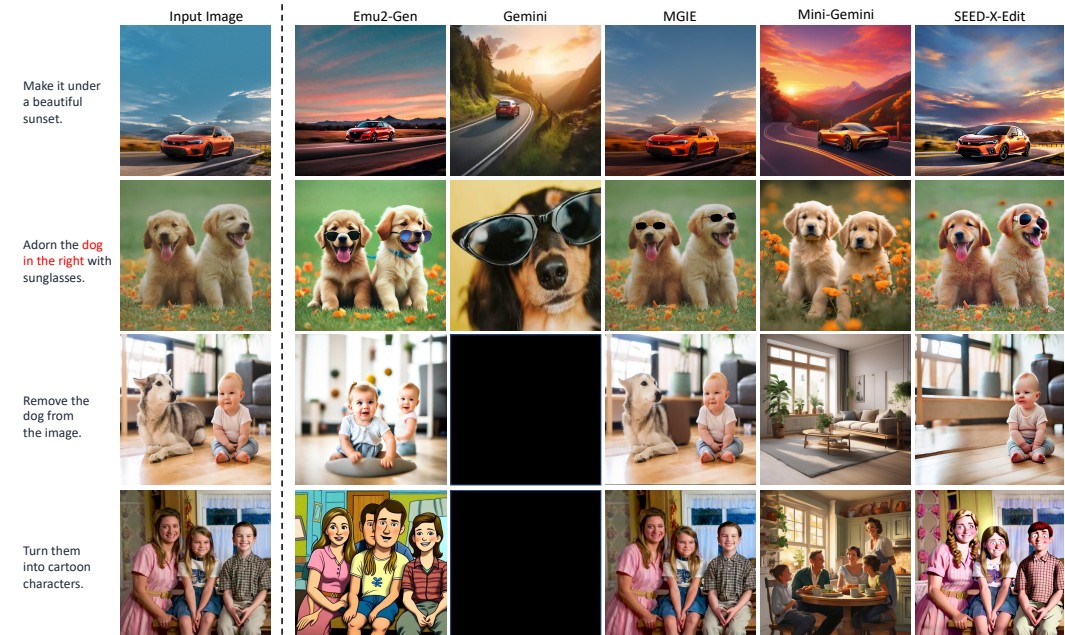

Figure 9: Qualitative comparison between MLLMs for image manipulation. SEED-X-Edit shows enhanced ability in adhering to instructions while preserving low-level details of input images. The black images result from Gemini's inability to display human images.

due to its failure to display images related to human portraits. Mini-Gemini generates **text prompts** as the input of a pre-trained SDXL model, which can not preserve the visual details of the input image. The examples demonstrate the effectiveness of our instruction model for high-precision image manipulation. Our MLLM accurately predicts visual semantic representations based on an input image and a language instruction, which serve as input for the U-Net. The visual de-tokenizer can further condition on the input image, ensuring the preservation of fine-grained details in the decoded images.

**Multimodal Comprehension** We provide qualitative examples of multimodal comprehension by SEED-X-I in Fig. 10 and Fig. 11. SEED-X-I can realize fine-grained object detection and perception, text-rich comprehension, fundamental mathematical computation, world-knowledge and common-sense reasoning, diagram understanding, which are crucial capabilities for its application in real-world scenarios.

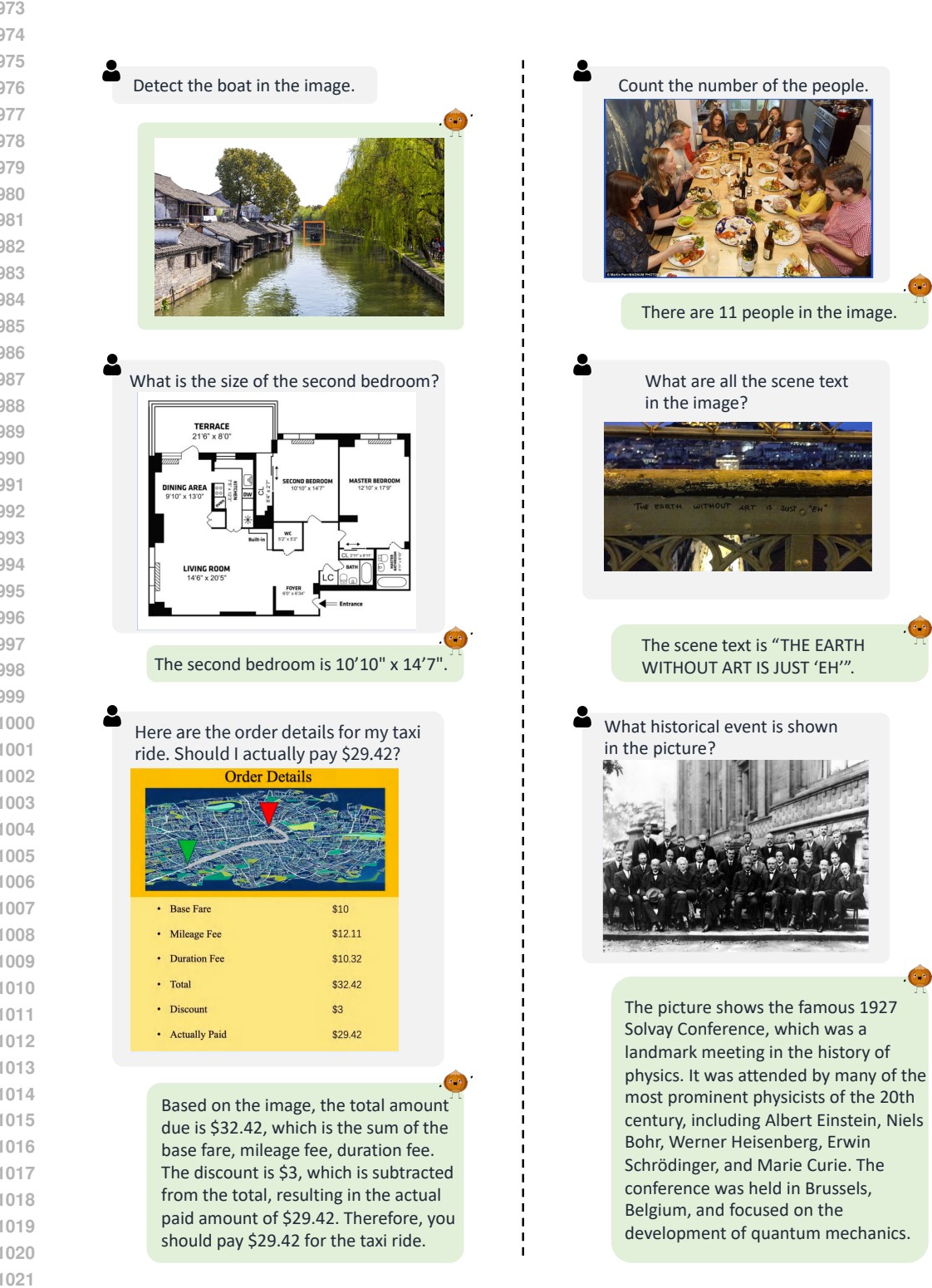

Figure 10: Qualitative examples of multimodal comprehension by SEED-X-I.

Can this be true in the real world?

Please explain the method in details.

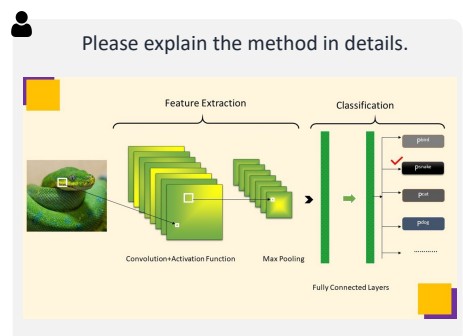

No, this is not possible in the real world. The image shows a stack of dice that are not physically connected, yet they are arranged in a way that suggests they are connected. This is an optical illusion, and the dice are not actually connected.

The method in the image is a Convolutional Neural Network (CNN) used for image classification. The process starts with the input image on the left, which is a green snake. This input goes through a convolutional layer, indicated by the green arrow pointing downwards. This convolutional layer is followed by a max pooling layer. The output of the max pooling layer feeds into a fully connected layer. This fully connected layer leads to the output of the network, which is the classification of the input image. The final output of the classification step is the classification of the input image, which is a green snake.

What is the meaning of the meme?

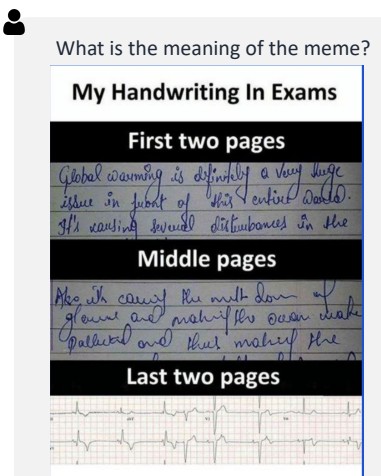

Is there anything in the image that can protect me from catching the flu virus when I go out?

Yes, the black face mask located at green bbox can help protect you from catching the flu virus when you go out.

The meme is comparing the quality of handwriting in the first two pages of an exam and the last two pages. It suggests that the quality of handwriting often deteriorates as the exam progresses. This is a humorous way to express the common experience of writing neatly at the beginning of an exam and becoming less neat as the time progresses.

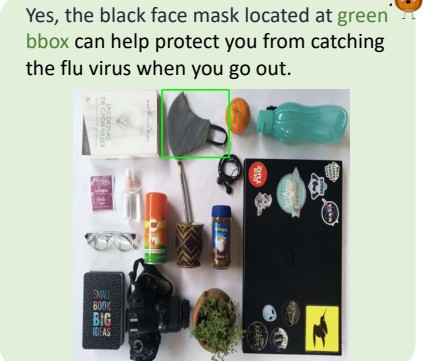

Figure 11: Qualitative examples of multimodal comprehension by SEED-X-I.

