# OpenReview forum: "SEED-X: Multimodal Models in Real World"
_ICLR.cc/2025/Conference — Submitted to ICLR 2025_

### Official Review · Reviewer_CjSH · 2024-10-30

**Soundness:** 2
**Presentation:** 2
**Contribution:** 3
**Rating:** 5
**Confidence:** 3

**Summary:**

The authors present a unified and versatile foundation model, namely, SEED-X. SEED-X integrates two features: (1) comprehending images of arbitrary sizes and ratios, and (2) enabling multi-granularity image generation.

**Strengths:**

The authors present a unified and versatile foundation model, namely, SEED-X. SEED-X integrates two features: (1) comprehending images of arbitrary sizes and ratios, and (2) enabling multi-granularity image generation.

**Weaknesses:**

1. The authors claim their results in Figure 1 and Figure 5 come from a "unified and versatile foundation model," but it seems these results are from different instruction-tuned models (SEED-X-Edit, SEED-X-PPT, etc.). This could mislead readers into thinking a single model handles all functionalities.
2. The authors only compare their approach with a few related works.
3. Results are only provided for MMB, SEED-Bench-2, and MME. Other benchmarks like VQA, MM-Vet, MMMUv, and MathVista are missing.
4. The model is pre-trained on LLaMA2, which is outdated. This raises concerns about the results being state-of-the-art, especially compared to newer models like LLaMA 3.2-Vision.

**Questions:**

1. For Figure 1 and Figure 5, the author claims that the results presented come from a "unified and versatile foundation model," but also mentions "after instruction tuning." I am curious whether the results of these different functionalities come from the series of instruction-tuned models mentioned in Sec 3.3.2, including SEED-X-Edit, SEED-X-PPT, SEED-X-Story, and SEED-X-Try-on. If they do indeed come from these different models, I believe the author's phrasing is attempting to mislead readers into thinking that these results come from a single "unified and versatile foundation model."
2. The term "MMB Single" in Table 2 refers to selecting questions from MMBench containing only one image? Why was this experimental setup chosen? The classification method I found in the original MMBench paper seems to be Overall, CP, FP-S, FP-C, AR, LR, RR. How do Seed-X and Seed-X-I perform on these subclasses compared to SOTAs?
3. In Table 2, it seems that comparisons are made with only a limited number of related works. Many of the related works mentioned in Table 1 and the results tested on benchmarks like MMBench have not been compared. What is the reason for not comparing with these works?
4. It seems that the authors only validated their approach on MMB, SEED-Bench-2, and MME, but what about the results on other benchmarks such as VQA, MM-Vet, MMMUv, MathVista, etc?
5. The authors pre-trained from LLaMA2-Chat-13B using LoRA, but LLaMA2 is already somewhat outdated, which raises concerns about whether the results are truly state-of-the-art. For example, how competitive is Seed-X, which is based on LLaMA2 pre-training, when compared to models like LLaMA 3.2-Vision?

---

### Official Review · Reviewer_f1AY · 2024-10-31

**Soundness:** 3
**Presentation:** 3
**Contribution:** 2
**Rating:** 6
**Confidence:** 4

**Summary:**

This paper introduces SEED-X, a multimodal base model that aims to improve the applicability of models in real-world applications by unifying multi-granularity understanding and generation capabilities. SEED-X introduces two key features: understanding images of arbitrary sizes and scales, and supporting multi-granularity image generation (including high-level instruction image generation and low-level image manipulation tasks). The paper shows the competitiveness of SEED-X on public benchmarks and demonstrates its effectiveness in multiple real-world applications.

**Strengths:**

1) The paper has a clear structure and coherent logic, and the ideas and thoughts are clearly presented through diagrams.
2) SEED-X receives and outputs arbitrary-size images, making it more useful in real open-world scenarios.
3) Compared to current MLLM methods, SEED-X integrates comprehension and generation ability simultaneously, e.g. detection, dynamic res img input, image gen, and high-precision editing.  Make it Generalist actually

**Weaknesses:**

1） Lack of quantitative ablation study, in Sec4.3 ablation study, the authors perform ablation studies on the training of visual de-tokenizer and the pre-training of SEED-X  and visualize the results of the ablation study. However, they lack the quantitative ablation study of their framework, e.g.: the image gridding operation which is claimed to support arbitrary size and aspect ratios. This operation contributes to one of their motivations, but lacks necessary quantitative analysis.

2) As shown in Tab.2 the experiment results compared to other MLLM methods on MLLM benchmarks are not competitive.

**Questions:**

1 As a general MLLM, the training and inference cost is essential, since the proposed model composes more functionality than other models, these details need to be clarified.

2 This paper conducts experiments on MME. MMB and SEED benchmarks. How’s its performance on general tasks for MLLM like VQA and cross-modal retrieval? The experiments did not show its general performance on MLLM tasks

---

### Official Review · Reviewer_69AZ · 2024-11-05

**Soundness:** 3
**Presentation:** 3
**Contribution:** 3
**Rating:** 5
**Confidence:** 5

**Summary:**

The authors claim that they present a framework named SEED-X that integrates image comprehension and generation capabilities. SEED-X demonstrates promising applications for real-world scenarios, supporting comprehension and generation with arbitrary sizes and aspect ratios via visual tokenization/de-tokenization and dynamic resolution encoding.

**Strengths:**

1.This paper introduces visual tokenization and de-tokenization method to support image generation and high-precision image manipulation.

2.This paper proposes dynamic resolution image encoding module which allows for the processing of images with various resolutions, enhancing the model’s adaptability to diverse real-world applications.

3.The proposed method integrates the image comprehension and generation into a single foundation model, which can be applicable in realworld scenarios.

**Weaknesses:**

1. Grammar Mistake: This paper is filled with numerous grammatical mistakes, which severely compromise its quality. Here are some examples.Ep1: The use of singular and plural forms of “work” is in chaos. In Line 81, it has “Some pioneering work” while in Line 99, it owns “none of the previous works”.Ep2: The form of verb has not been used correctly. Such as in Line 209, “which effectively incorporate the aforementioned characteristics for real-world applications ...” should be changed as “which effectively incorporates the aforementioned characteristics for real-world applications ...”

2. Missing Component: I couldn’t find the summary of the authors’ contributions, which usually can be found in the end of the Introduction section. It makes me confused about the authors’ contributions.

3. Unknown Structure: In this paper, the authors only depict the independent modules in the method, while lacking of the comprehensive and total processing pipeline of the method. For example, the authors may provide with an architecture figure about their method and introduce the processing pipeline about image comprehension and generation step by step, which can be better for the readers to understand their method.

4. Extra Experiment: In the ablation study, 1) the authors only provide the visualization results without numerical metrics which could better reflect the comprehensive performance. 2). the authors only conduct the experiments belong to the image generation, while image comprehension is also significant in their settings. 3). the authors only ablate the number of learnable queries, which can not reflect the effectiveness of their proposed modules: visual tokenization and de-tokenization, dynamic resolution image encoding.

5. Insufficient Innovation: I concern about the innovations about this paper from the following perspectives: 1). The two proposed innovations are disconnected from each other, and there is no strong correlation in the paper. 2). The authors propose a grand blueprint, while their innovations look like a bit ordinary, especially considering about their moderate performance compared with GPT.

**Questions:**

Please kindly refer to the weakness.

**Details Of Ethics Concerns:**

Please kindly refer to the weakness.

---

### Official Review · Reviewer_WNpz · 2024-11-08

**Soundness:** 3
**Presentation:** 3
**Contribution:** 2
**Rating:** 5
**Confidence:** 4

**Summary:**

The paper presents SEED-X, an advanced multimodal foundation model designed for enhanced real-world applicability in both comprehension and generation across diverse user inputs. SEED-X builds upon prior work, SEED-LLaMA, and addresses two main challenges: (1) understanding images of varying sizes and aspect ratios and (2) facilitating multi-granularity image generation. These features enable SEED-X to handle high-level creative generation tasks and precise image manipulation. The model incorporates a visual tokenizer and a novel de-tokenization approach, enhancing image fidelity and allowing for detailed editing based on conditional inputs. SEED-X also supports dynamic resolution image encoding, enabling seamless processing of images with arbitrary dimensions without compromising visual details. SEED-X was pre-trained on extensive multimodal data and underwent instruction tuning across various domains, resulting in specialized versions such as SEED-X-Edit, SEED-X-PPT, and SEED-X-Story, each tailored for specific applications. In evaluations, SEED-X achieved competitive results in both comprehension and generation benchmarks, demonstrating its robustness across multimodal large language model (MLLM) benchmarks.

**Strengths:**

1. This paper tries to solve a very interesting and fundamental problem, a vision-language multimodal foundation model that enables understanding and generation, and shows a bunch of very interesting application scenarios after instruction tuning like image editing, comprehending and generation.

2. While not entirely optimal, the proposed designs, including the visual tokenizer and the image encoding of dynamic resolution are both reasonable.

3. For the visual representation, currently SEED-X leverages the continuous tokens predicted from the learnable queries and optimized with regression loss. How about using discrete tokens and next-word prediction objectives? It seems like this way will better unify the language and the image representation. More discussions could also be incorporated here.

**Weaknesses:**

1. The Dynamic Resolution Image Encoding section is very interesting. However, the current processing way still inherits some drawbacks, for example we still need to concat the features of all larger patches and resized global image, which will inevitably result in the increased sequence length and contain redundant information. It is also important to show more ablation studies here to show its more significance.

2. What is the motivation of the visual de-tokenizer training in the second stage and how will it help? As shown in Figure 3,  SEED-X uses the image editing data to further finetune the visual detokenizer, hence the conditional image has some differences with the reconstruction target, for example, the removed dog shown in the example. I partially agree with the claim of Section 3.1 that the given reference could help recover more fine-grained details, but somehow the editing capability which should be fully provided by the large multimodal model actually came from the UNET of SD-XL to some extent, since most of the time we only want the tokenizer to compress the signal rather than modify the signal. Could the authors share any ablation study to see how it will impact the large model capability?

3. During inference, when the model wants to generate a new image I wonder if the previous image in this sequence is also required to send to the de-tokenizer as a condition?

4. In Figure 4, may I ask what is the white token (between IMG and the regressed image features)?

5. The writing of the paper could be improved as some parts are very confusing and high-level, I would recommend to include more details regarding the implementation and the training .

**Questions:**

See the weakness

---

### Official Review · Reviewer_Agua · 2024-11-09

**Soundness:** 2
**Presentation:** 2
**Contribution:** 2
**Rating:** 3
**Confidence:** 5

**Summary:**

This paper presents SEED-X, a VLM that is designed for both multimodal understanding and AIGC. The key idea is to leverage a pretrained query-based visual decoder and inject it into the VLM for image generation and understanding training at the same time. The experiments show that SEED-X achieves promising results on both image generation and understanding.

**Strengths:**

- The problem of pursuing joint image understanding and generation with LLMs seems new and relevant, which would be of value to both the community and the industry.
- The method is pretty simple, but it might not be easy to follow as a visual decoder has to be pretrained.
- The results look promising.

**Weaknesses:**

- **Novelty**. The proposed method looks like a combination of Emu which regresses continuous visual tokens and DreamLLM which uses continuous queries as image generation conditions. However, I am not saying the efforts of such exploration should not be encouraged.
- **Writing**. In the method part, it is quite unclear about every design's motivation. There is not a single reference in this part, which may lead to confusion on method detail and differences from previous works. The authors should carefully discuss the technical motivation and the literature comparison.
- **Evaluation**. Current experiments only consist of MMBench and MME. More classical benchmarks like typical VQA including VQAv2 and advanced benchmarks like MM-Vet should be used.

**Questions:**

Can SEED-X be used for video understanding? For example, compare with VILA-U on MSVD-QA and TGIF-QA.

---

### Meta-Review · Area_Chair_LV7H · 2024-12-19

**Metareview:**

This paper presents SEED-X a multimodal model that (1) comprehends images of arbitrary sizes and ratios, and (2) enables multi-granularity image generation. The paper received scores of 3,5,5,6,5.  The reviewers found some aspects of the proposed problem and approach interesting.  However, the critical issues that were raised include limited novelty, insufficient experiments, and issues with clarity.  However, there was no rebuttal.  The AC agrees with the reviewers' concerns, and recommends reject.

**Additional Comments On Reviewer Discussion:**

No rebuttal provided, and there was no further discussion between the reviewers and AC.

---

### Decision · Program_Chairs · 2025-01-22

Reject